# A global grid model for the estimation of zenith tropospheric delay considering the variations at different altitudes

Liangke Huang[1,2], Shengwei Lan[1,2], Ge Zhu[3], Fade Chen[1,2], Junyu Li[1,2], Lilong Liu[1,2]

[1] College of Geomatics and Geoinformation, Guilin University of Technology, Guilin, 541006, China.

[2] Guangxi Key Laboratory of Spatial Information and Geomatics, Guilin, 541006, China.

[3] College of Surveying and Geo-informatics, Tongji University, Shanghai, 200092, China.

*Correspondence to*: Ge Zhu (zhuge@tongji.edu.cn)

**Abstract.** The accuracy of tropospheric delay correction heavily depends on the quality of the tropospheric model, and zenith tropospheric delay (ZTD) is an important factor affecting the tropospheric delay. Therefore, it is essential to establish a precise ZTD empirical model. The existing ZTD models have limitations such as using a single fitting function, neglecting daily cycle variations, and relying on only one resolution grid data for modeling. To address these limitations, we proposed a global piecewise ZTD empirical grid (GGZTD-P) model. This model considers the daily-cycle variation and latitude factor of ZTD, using the sliding window algorithm based on fifth-generation European Centre for Medium Range Weather Forecasts (ERA5) atmospheric reanalysis data. The ZTD data from 545 radiosonde stations and the second Modern-Era Retrospective analysis for Research and Applications (MERRA-2) atmospheric reanalysis data are used to validate the accuracy of the GGZTD-P model. The results indicate that the GGZTD-P model outperforms the GPT3 model, exhibiting 26% and 53% lower bias and RMS, respectively, when using radiosonde stations as reference values. Furthermore, when evaluated using MERRA-2 atmospheric reanalysis data, the GGZTD-P model consistently exhibits superior performance across various latitude regions. It's expected that application of this new model will provide improved services for high-precision GNSS positioning and GNSS meteorology.

## 1 Introduction

Global Navigation Satellite System (GNSS) signals experience time delay and bending effects as they pass through the neutral atmosphere, resulting in tropospheric delay. Depending on the satellite's altitude angle, tropospheric delay ranges from 2-20 meters in the zenith direction (Li et al., 2018; Yao et al., 2018; Yao et al., 2019; Li et al., 2021), significantly affecting positioning accuracy. Accurate Zenith Tropospheric Delay (ZTD) information can improve GNSS positioning precision (Nafisi et al., 2012; Zhang et al., 2022; Zhao et al., 2023a; Zhang et al., 2020; Zhou et al., 2021). Currently, ZTD models can be divided into two categories: the one requires real-time measurement of meteorological parameters, the other is empirical models fitted according to a large volume of data and do not require meteorological parameters.

Hopfield (1969) proposed the use of radiosonde data to establish the global tropospheric delay model, known as the Hopfield model. This model requires the input of temperature, pressure, water vapor pressure, and station location to calculate tropospheric data. Saastamoinen (1972) further divided the troposphere into two profiles using the standard atmospheric model of the United States and calculated other parameters of tropospheric delay indirectly

with meteorological parameters to obtain ZTD. Based on Hopfield's work, Black (1978) refined the Hopfield model and established the famous Black model. These models provide high-precision tropospheric data through measured meteorological parameters. However, not all GNSS stations are equipped with expensive meteorological sensors, limiting the availability of real-time meteorological data and hindering the use of these models.

To overcome this limitation, researchers have developed several empirical models that do not rely on measured meteorological parameters (Li et al., 2020; Zhang et al., 2021). Leandro et al. (2006; 2008) developed the UNB series model, and Penna et al. (2001) developed the EGNOS model. Krueger et al. (2004) utilized the NCEP atmospheric reanalysis data to establish the TropGrid model, which has a horizontal resolution of 1°×1° and provides 25% greater accuracy than the EGNOS model globally. Based on the TropGrid model, Schüler (2014) established the TropGrid2 model by taking into account the daily cycle variation of ZTD using multi-year Global Data Assimilation System (GDAS) data, which improves the time resolution but ignores the semi-annual cycle variation of ZTD. The (Global Pressure and Temperature) GPT series models (Böhm et al., 2007; Lagler et al., 2013; Böhm et al., 2015; Landskron et al., 2018) are based on European medium-term prediction center (ECMWF) atmospheric reanalysis data and consider the temperature and pressure in cycles. Lagler et al. (2013) constructed the GPT2 model based on the GPT model, using 10-year ERA-Interim data with a resolution of 5°×5°. Böhm et al. (2015) further improved the GPT2 model to obtain the GPT2w model, which is currently recognized as the ZTD model with high accuracy. The latest update to the GPT2w model is the GPT3 model, which only modifies the empirical mapping function (Sun et al., 2019; Ding et al.,2020), as compared to the GPT2w model. In order to address the limitations of current ZTD models, researchers have proposed the use of a sliding window algorithm (Huang et al., 2019; Huang et al., 2021) to construct models with different window sizes. This approach can further optimize the model parameters. Furthermore, Yang et al. (2021) employed an Artificial Neural Network (ANN) to effectively mitigate the systematic deviation within the GPT3 model, leading to improved ZTD accuracy in Hong Kong, China. Zhao et al. (2023) took into account the residual term between the GPT3 model and GNSS observations ZTD to develop a novel model specific to China (CHZ). Additionally, Li et al. (2023) discover the disparities between ERA5 and GNSS-based ZTD, prompting the creation of a new global model (IGGZTD-S). This new model demonstrated exceptional performance in Precise Point Positioning (PPP), particularly in the vertical direction.

This paper proposes a new global piecewise ZTD empirical grid model called GGZTD-P derived from the established GZTD-P vertical adjustment model (Zhu et al., 2022). The GGZTD-P model takes into account the fine daily variation of ZTD and latitude factors to provide a more accurate representation of the atmosphere. The accuracy of the GGZTD-P model was evaluated by comparing it with profiled ZTD data from 545 radiosonde stations in 2017 and 2018, as well as the Modern-Era Retrospective analysis for Research and Applications, Version 2 (MERRA-2) atmospheric reanalysis data from 2017. It should be explained that the ZTD data of radiosonde and MERRA-2 is calculated by integration. The results were also compared with the GPT3 model to assess the performance of the GGZTD-P model. The aim of this study is to provide more important reference for GNSS meteorology and positioning.

## 2. Data and methods

### 2.1 Atmospheric reanalysis data

The fifth generation European Centre for Medium Range Weather Forecasts (ERA5) atmospheric reanalysis data provides tropospheric parameters such as temperature, pressure, and humidity with a high spatial resolution of 0.25°×0.25° (latitude × longitude) and a temporal resolution of 1 hour. The ERA5 data is recognized as a valuable resource for research and applications in GNSS meteorology and positioning (Chen et al., 2021; Prado et al., 2022; Sun et al., 2023).

MERRA-2 is a state-of-the-art atmospheric reanalysis dataset developed by NASA (Chen et al., 2019; Huang et al., 2022; Randles et al.,2017). It represents a major advancement over its predecessor, MERRA, as it incorporates aerosol observations from space and their interactions with physical processes (Gupta et al., 2020;Huang et al., 2020; Zhao et al., 2022). MERRA-2 provides a wealth of surface and profile meteorological parameters. The data are distributed across 42 profiles according to standard atmospheric pressure. The surface parameters, such as surface pressure, surface temperature, specific humidity, and surface elevation have a temporal resolution of 1 hour and a spatial resolution of 0.5°×0.625° (latitude × longitude). The profile parameters, such as temperature, specific humidity, and high potential, have a temporal resolution of 6 hours and a spatial resolution of 0.5°×0.625°.

### 2.2 Radiosonde data

Sounding balloons are typically launched twice daily at 00:00 UTC and 12:00 UTC, and collect meteorological vertical profile information such as pressure, temperature, and relative humidity at specific pressure levels. Radiosonde data offers precise meteorological observations acquired through direct measurements. Zhao et al. (2019) found that ZTD derived from radiosonde is validated using GNSS data, with RMS errors of 19.1 mm. Shangguan et al. (2022) discovered that the bias and RMS of the ZTD data from 180 radiosonde stations compared with data from ERA5 worldwide were 8.5 mm and 13.2 mm, respectively. Radiosonde data are widely used to evaluate the precision of other atmospheric reanalysis data or tropospheric parameter models (Tang et al., 2013; Zhou et al., 2017; Bonafoni et al., 2019).

### 2.3 Calculation principle and methodology

Atmospheric reanalysis data can provide meteorological parameters according to standard atmospheric pressure profiles. Integration method is used to calculate the ZTD. First, the atmospheric refractivity index is calculated using the meteorological parameters of each profile. Next, the refractive index is integrated at the height of each profile to obtain the vertical profile information of ZTD at each grid point. Finally, by hierarchically combining the ZTD information obtained from the integration, the vertical profile information of ZTD at each grid point can be obtained. The integral formula used is as follows (Thayer, 1974):

$$N = k_1 \frac{(P-e)}{T} + k_2 \frac{e}{T} + k_3 \frac{e}{T^2} \tag{1}$$

$$e = \frac{Sh \cdot P}{0.622} \tag{2}$$

$$ZTD = 10^{-6} \int_{h_L}^{h_{top}} N\mathrm{d}H \qquad (3)$$

where, $N$ stands for the total atmospheric refractivity, $P$ stands for the atmospheric pressure (hPa), e stands for the water vapor pressure (hPa), $Sh$ stands for the specific humidity, $T$ stands for the temperature, $h$ stands for the elevation, $h_L$ stands for the height at the bottom of the atmospheric data integration calculation, and $h_{top}$ stands for the height at the top of the atmospheric data integration calculation. $k_1 = 77.604\mathrm{K/Pa}$, $k_2 = 64.79\mathrm{K/Pa}$, $k_2' = 22.97\mathrm{K/hPa}$ and $k_3 = 375463\mathrm{K^2/hPa}$ are all constant coefficients.

### 3. characteristic analysis

### 3.1 Temporal characteristic analysis

To construct a high-precision global ZTD grid model, it is essential to analyze the spatiotemporal characteristics of ZTD over the globe. Six representative ERA5 grid sites data, distributed evenly around the world, are used to calculated the average daily ZTD time series for each site from 2012-2016. These time series are then fitted using cosine and sine functions with annual and semi-annual periods. The results are presented in Figs. 1 and 2.

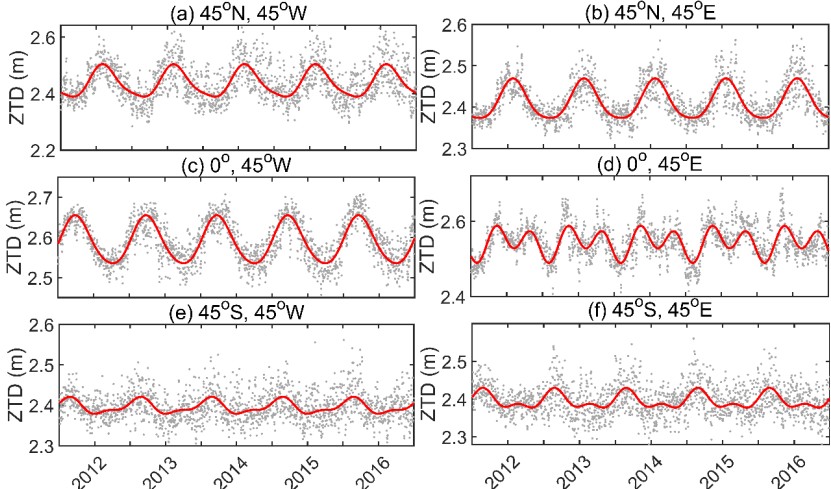

**Figure 1. Average daily ZTD time series from 2012 to 2016 for 6 ERA5 grid sites distributed in each hemisphere of the world.**

In Fig. 1, the gray points represent the daily mean ZTD, while the red lines represent the fitted values. The figure reveals that ZTD exhibits a global fluctuation range of 2.2-2.7 m, with values ranging from 2.3-2.6 m in the Northern and Southern hemispheres, and 2.4-2.7 m near the equator. Notably, ZTD shows significant seasonal variations with large fluctuations in the Northern hemisphere and near the equator. Conversely, ZTD shows a smaller fluctuation range in the southern hemisphere with no apparent seasonal variations.

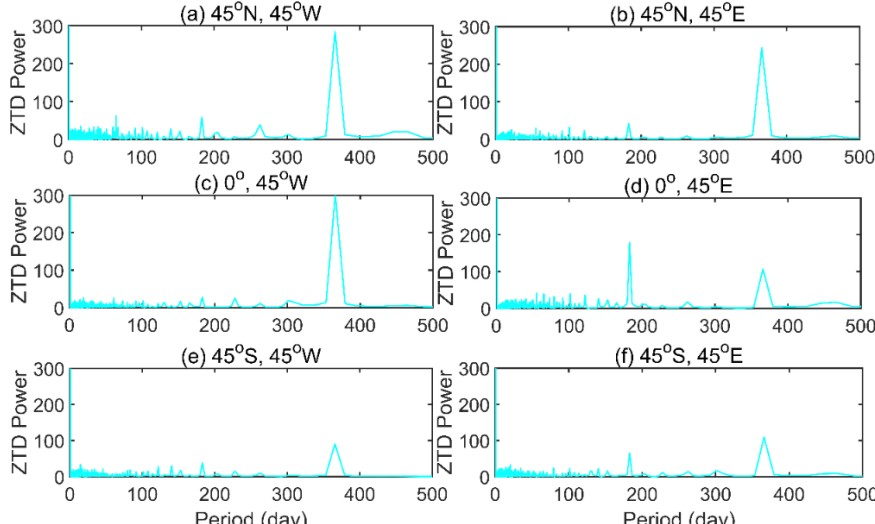

Figure 2. ZTD fast Fourier transform spectrum analysis of 6 representative ERA5 grid points.

In Fig. 2, the Fast Fourier Transform (FFT) spectrum analysis was performed on six selected ERA5 grid points, uniformly distributed in each hemisphere of the globe. The results indicate that the highest annual period power of ZTD is observed in the Northern hemisphere, accompanied by notable annual and semi-annual period variations. In contrast, the Southern hemisphere shows a lower annual period power but exhibits significant annual and semi-annual period variations. Near the equator in the Western hemisphere, the annual period power is greater, displaying clear annual period variations. However, the semi-annual period power is lower, indicating inconspicuous semi-annual period variations. In the Eastern hemisphere at the equator, the semi-annual period power is higher than the annual period power, indicating clear annual and semi-annual period variations.

To further confirm the daily period variations of ZTD, six ERA5 atmospheric reanalysis data grid points are selected randomly for on January 1, 2015. The results are presented in Fig. 3.

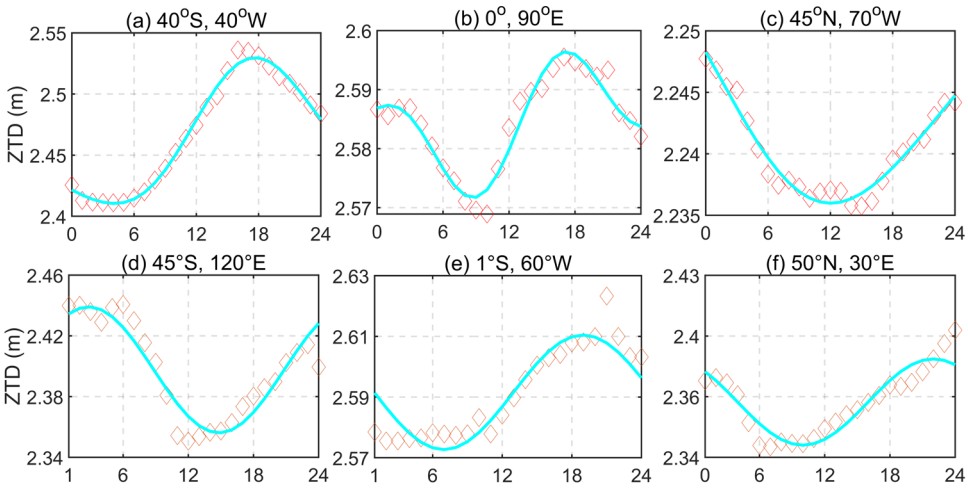

Figure 3. Time series of daily variations of ZTD.


Fig. 3 reveals that ZTD exhibits significant daily period variations in the six selected grid points, particularly at the grid points (0°, 90°E) and (1°S, 60°W) where significant daily period characteristics are observed. Thus, when constructing global ZTD models, it is important to consider daily period variations.

**3.2. Spatial characteristic analysis**

To analyze the global distribution of ZTD, the average daily ZTD surface information of ERA5 atmospheric reanalysis data is calculated for the year 2015. The results are presented in Figs. 4 and 5.

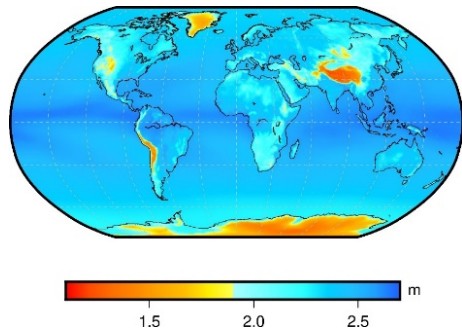

**Figure 4. Average daily surface ZTD values for ERA5.**


Fig. 4 reveals that ZTD has smaller values in the Arctic Ocean region, western regions of North and South America, Antarctica, southeastern Africa, and Asia. This may be due to the undulating terrain and higher altitude in these areas, resulting in lower ZTD values. The lowest values were found in Asia and Antarctica.

To further verify the global distribution characteristics of ZTD, including the annual mean, annual period
amplitude, semi-annual period amplitude, daily period amplitude, and semi-daily period amplitude, the results are presented in Fig. 5.

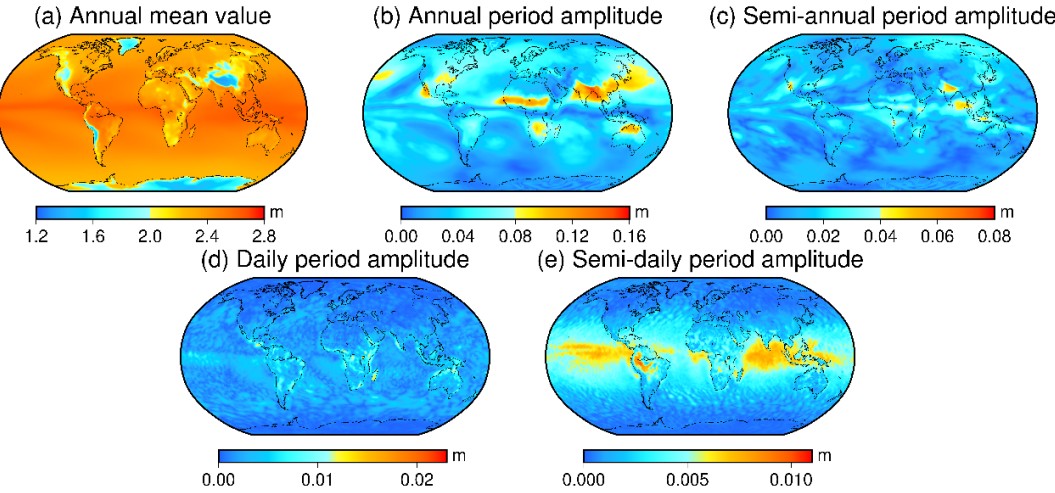

**Figure 5. Distribution characteristics of annual mean, annual period amplitude, semi-annual period amplitude, daily period amplitude and semi-daily period amplitude of ZTD.**


Fig. 5 reveals that ZTD has a large annual mean value in the low-latitude region of the world, and a small annual mean value in Antarctica, China, Arctic Ocean, and the western regions of North and South America. The amplitude of the annual period is significant in southern Asia and central Africa, with the northern hemisphere showing a more obvious annual periodicity amplitude than the southern hemisphere. Northeast Asia, Oceania, southern Africa, and

North America show obvious amplitude of semi-annual period. For the daily amplitude of ZTD, significant variations are observed in the low latitudes, particularly in South America, Africa, and Oceania, whereas the high latitudes show less prominent daily amplitude. Similarly, near the equator, a significant semi-daily period amplitude is observed, particularly in the northern region of the Pacific, South America, and the Indian Ocean, where the largest semi-daily period amplitude is observed. This may be due to the fact that these regions are located at the junction of

the ocean and land and are in the same direction as the northeast (Northern Hemisphere) and southeast (Southern hemisphere) equatorial trade winds (Yao et al., 2012), indicating that the distribution of ZTD is not only related to meteorological variables and topography, but also influenced by thermodynamic circulation (Yao et al., 2013). The low and mid-latitude regions in the world have more apparent semi-daily period amplitude, whereas the high latitudes of the world show a less obvious amplitude of the semi-daily period.

According to relevant studies, ZTD values are primarily associated with latitude factors on a global scale, while showing a smaller correlation with longitude factors (Chen et al., 2020; Huang et al., 2022). In order to further verify the distribution of ZTD values on a global scale, under the condition of controlling variables, the hierarchical ZTD vertical profile information of ERA5 atmospheric reanalysis data at 00:00 UTC on January 1, 2015, is used to interpolate the ZTD values of each grid points at the height of 6 km. The results are shown in Fig. 6.

Fig. 6 illustrates the global distribution of ZTD values obtained from the ERA5 atmospheric reanalysis data. The analysis reveals a strong correlation between ZTD values and latitude factors. Specifically, ZTD values tend to be lower in high-latitude regions and higher in middle and low-latitude regions. The smallest ZTD values are observed in the northeast region of North America and the Arctic Ocean region. On the other hand, the global distribution of ZTD values has minimal correlation with longitude factors. As a result, when developing the empirical grid model

for global ZTD, the impact of latitude on model accuracy should be taken into account.

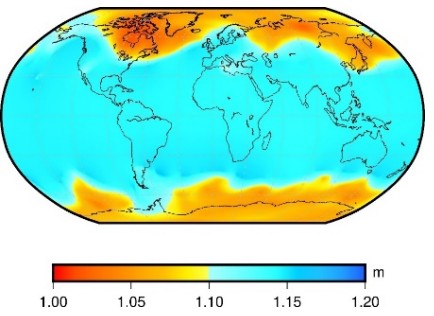

**Figure 6. Global distribution map of ZTD value of ERA5 data at 6 km height at 0:00 UTC on January 1, 2015.**

## 3.3 Construction of global piecewise ZTD vertical correction grid model

In order to optimize the model parameters, the grid was divided into a horizontal resolution of 2°×2°. A piecewise function was employed to accurately model each elevation interval of the grid, while also accounting for seasonal variations in the model. As a result, the global piecewise ZTD vertical profile model (GZTD-P) was developed, and it can be expressed by the following formula:

$$ZTD_t = \begin{cases} ZTD_{r1} * \exp\left(H_{s1} * \left(H_t - H_r\right)\right) & \left(H_t < 3\text{km}\right) \\ ZTD_{r2} * \exp\left(H_{s2} * \left(H_t - H_r\right)\right) & \left(3\text{km} \leq H_t < 8\text{km}\right) \\ ZTD_{r3} * \exp\left(H_{s3} * \left(H_t - H_r\right)\right) & \left(8\text{km} \leq H_t < 16\text{km}\right) \\ ZTD_{r4} * \exp\left(H_{s4} * \left(H_t - H_r\right)\right) & \left(H_t \geq 16\text{km}\right) \end{cases} \tag{4}$$

$$\begin{aligned} H_s = {} & \alpha_1 + \alpha_2 \cdot \cos\left(2\pi \frac{DOY}{365.25}\right) + \alpha_3 \cdot \sin\left(2\pi \frac{DOY}{365.25}\right) \\ & + \alpha_4 \cdot \cos\left(4\pi \frac{DOY}{365.25}\right) + \alpha_5 \cdot \sin\left(4\pi \frac{DOY}{365.25}\right) \end{aligned} \tag{5}$$

In Eqs. (4) and (5), $H_s$ stands for ZTD value at the average elevation, $H_t$ stands for target elevation, $H_r$ stands for reference elevation, and $ZTD_t$ stands for ZTD value at target elevation. $a_i$ stands for the constant, annual and semi-annual period correction factor. $ZTD_{r1}$, $ZTD_{r2}$, $ZTD_{r3}$, $ZTD_{r4}$ stands for ZTD values at the reference elevations of different piecewise, respectively.

## 3.4 Construction of global piecewise ZTD empirical grid model

ERA5 atmospheric reanalysis data ZTD on the surface will be uniformly converted to the position of the sliding window's average elevation. This conversion is based on the piecewise global ZTD vertical profile model, GZTD-P model, taking into account the elevation position of each window. The model is based on the ZTD values at the sliding window's average elevation. Utilizing the GZTD-P model, ZTD data for all window from 2012 to 2016 were vertically interpolated to calculate the ZTD value at the average elevation of each window after correction. The detailed process is shown in Fig. 7. To estimate the coefficients in each window, the least-squares adjustment is utilized, considering the annual, semi-annual, daily, and semi-daily variations, as well as the latitude factor. Finally, the global ZTD empirical grid model (GGZTD-P) is developed based on a piecewise expression, with a resolution of 1°×1°. The model can be expressed as follows:

$$ZTD_t = \begin{cases} ZTD_r * \exp\left(H_{s1} * \left(H_t - H_r\right)\right) \left(H_t < 3\text{km}\right) \\ ZTD_3 * \exp\left(H_{s2} * \left(H_t - 3\right)\right) \left(3\text{km} \leq H_t < 8\text{km}\right) \\ ZTD_8 * \exp\left(H_{s3} * \left(H_t - 8\right)\right) \left(8\text{km} \leq H_t < 16\text{km}\right) \\ ZTD_{16} * \exp\left(H_{s4} * \left(H_t - 16\right)\right) \left(H_t \geq 16\text{km}\right) \end{cases} \tag{6}$$

$$MP = A_0 + A_1 \cdot \cos\left(2\pi \frac{HOD}{24}\right) + A_2 \cdot \sin\left(2\pi \frac{HOD}{24}\right)$$
$$+ A_3 \cdot \cos\left(4\pi \frac{HOD}{24}\right) + A_4 \cdot \sin\left(4\pi \frac{HOD}{24}\right)$$

(7)

$$A_i = \alpha_1 + \alpha_2 \cdot \varphi + \alpha_3 \cdot \cos\left(2\pi \frac{DOY}{365.25}\right) + \alpha_4 \cdot \sin\left(2\pi \frac{DOY}{365.25}\right)$$
$$+ \alpha_5 \cdot \cos\left(4\pi \frac{DOY}{365.25}\right) + \alpha_6 \cdot \sin\left(4\pi \frac{DOY}{365.25}\right)$$

(8)

In Eqs. (6) (7) and (8), $MP$ stands for the ZTD value at the average elevation, 3 km elevation, 8 km elevation and 16 km elevation, and $A_i$ stands for the daily period coefficient. $a_i$ stands for the constant, latitude, annual and semi-annual period correction factor, $\varphi$ stands for latitude, $DOY$ stands for year day, $HOD$ stands for time.

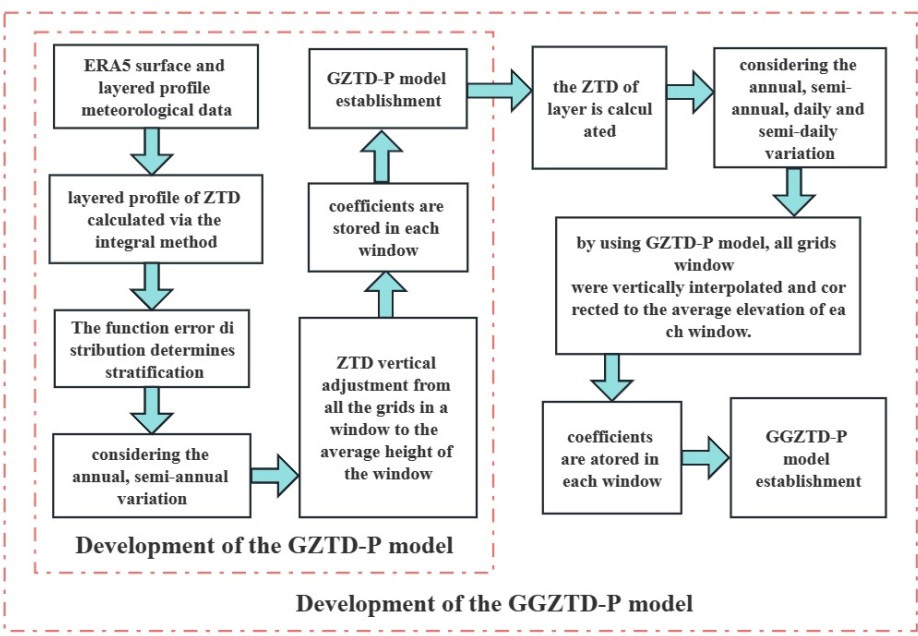

Figure 7. Flowchart depicting the development and use of the model.

The GGZTD-P model is easy to use. All that is required is the user's longitude and latitude, elevation, day of year, and hour of day to find the model coefficient closest to their position. The ZTD value at the average elevation can be corrected to the ZTD value at the target elevation using the formula provided. The GGZTD-P model can be used not only as an empirical grid model to estimate the ZTD value, but also as a ZTD vertical profile model for vertical interpolation.

**4 Accuracy verification**

In order to verify the stability of the established model in the global region, two sets of data are used as reference values and compared with the GPT3 model. The GPT3 model was developed utilizing a 15-year dataset of monthly average ERA-Interim profiles. Currently, it functions as a highly accurate tropospheric model.

$$M = S_0 + S_1 \cdot \cos\left(2\pi \frac{DOY}{365.25}\right) + S_2 \cdot \sin\left(2\pi \frac{DOY}{365.25}\right)$$
$$+ S_3 \cdot \cos\left(4\pi \frac{DOY}{365.25}\right) + S_4 \cdot \sin\left(4\pi \frac{DOY}{365.25}\right)$$

(9)

In Eq. (9), $M$ represents the tropospheric meteorological parameters (temperature, water vapor pressure, specific humidity, etc), and $S_i$ represents the annual mean value, annual and, semi-annual period coefficients. The Saastamoinen model and the Askne model were adopted to compute zenith hydrostatic delay (ZHD) and zenith wet delay (ZWD) with the obtained meteorological parameters.

$$ZHD = \frac{0.0022768P}{1 - 0.00266\cos(2\theta) - 0.00000028h}$$

(10)

$$ZWD = 10^{-6}(k_2' + \frac{k_3}{T_m}) \cdot \frac{R_d}{(\lambda + 1) \cdot g_m} \cdot e$$

(11)

In Eqs. (10) and (11), $P$ stands for pressure, $\theta$ stands for latitude, $h$ stands for elevation, $g_m$ is the average acceleration of gravity, $\lambda$ stands for the drop factor of water vapor pressure, $T_m$ stands for the atmospheric weighted mean temperature, and $k_2' = 22.97\text{K/hPa}$, $k_3 = 375463\text{K}^2/\text{hPa}$, $R_d = 287.054\text{J/kg} \cdot \text{K}$ are all constant coefficients.

**4.1 Radiosonde stations data was used for verification**

In this study, the accuracy of Zenith Total Delay (ZTD) obtained by the GGZTD-P model is compared with that of the GPT3 model. The ZTD layered profiles at 545 radiosonde stations in 2017 and 2018 are used as reference values. The accuracy of each model was statistically evaluated, as shown in Table 1. and Fig. 8.

Tab. 1 presents the results of the ZTD profile verification for global radiosonde stations, indicating the performance of the GPT3 and GGZTD-P models. Both models exhibit a positive average bias, implying that the ZTD values obtained by these models are generally higher than the ZTD values obtained from radiosonde stations. However, the average bias of the GGZTD-P model is 0.86 cm, which is 3.02 cm (78%) less than that of the GPT3 model. In terms of root-mean-square (RMS) error, the average RMS error of the GPT3 model is 6.84 cm, while the average RMS error of the GGZTD-P model is 3.23 cm, resulting in an accuracy improvement of 3.61 cm (53%) compared to the GPT3 model. The enhanced performance of the GGZTD-P model can be attributed to its ability to accurately simulate the variations of zenith tropospheric delay in the vertical direction through a piecewise fitting approach, which reduces the fitting error for each height interval. Overall, the GGZTD-P model demonstrates

excellent performance in validating the ZTD values of radiosonde stations, showing its superior accuracy and suitability for ZTD estimation.

**Table 1 The accuracy of GGZTD-P model and GPT3 model was verified using ZTD profiled data at radiosonde stations in 2017 and 2018.**

| Model | GGZTD-P | | GPT3 | |
|---|---|---|---|---|
| | Bias (cm) | RMS (cm) | Bias (cm) | RMS (cm) |
| Max | 3.21 | 13.60 | 7.83 | 14.37 |
| Min | -11.21 | 1.85 | -10.00 | 2.45 |
| Mean | 0.86 | 3.23 | 3.88 | 6.84 |

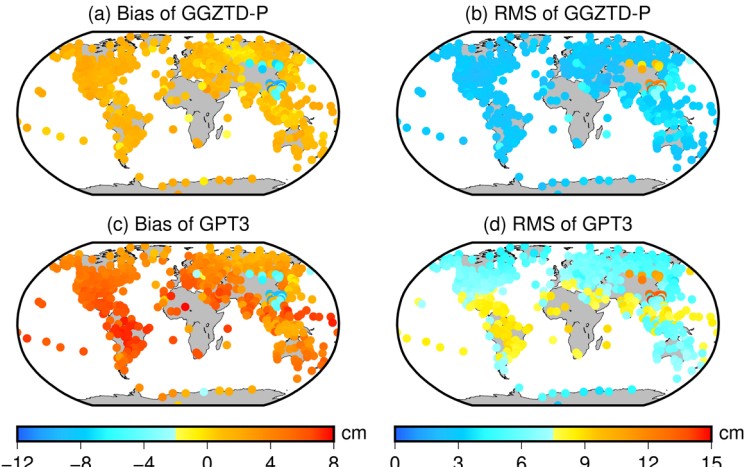

**Figure 8. ZTD profiled accuracy distribution of GGZTD-P model and GPT3 model for global radiosonde stations in 2017 and 2018.**

In Fig. 8, the ZTD profiled values of radiosonde stations calculated by the GGZTD-P model on a global scale mainly exhibit small positive bias, while those in the Asian region show large negative bias. This indicates that the calculated ZTD values by the GGZTD-P model are generally larger than the ZTD values of the radiosonde stations on a global scale, and smaller than the ZTD values in the Asian region. Similarly, the ZTD values of radiosonde stations calculated by the GPT3 model on a global scale exhibit mainly positive bias, with large negative bias values in Asia and large positive bias values in North America, South America, Africa, and the Pacific region. Both the GGZTD-P and GPT3 models show obvious negative bias values in the China region, suggesting that the ZTD estimated by the two models in this region is lower than the ZTD values of the radiosonde stations. This could be due to the complex and volatile climate and large topographic relief in China, which makes it difficult to accurately simulate the ZTD.

The GPT3 model shows smaller RMS values in North America, Europe, Antarctica, Oceania, and North Asia, but larger RMS values in the central and southern parts of Asia, especially in China, reflecting the largest RMS error.

This may be due to the more pronounced variations in terrain in the China region, making it challenging to accurately simulate the radiosonde stations. The GGZTD-P model shows a small RMS value globally, particularly in North America, South America, Europe, Oceania, and Antarctica, reflecting better correction accuracy. However, it exhibits a large RMS error in the Asian region for the reasons mentioned above. Compared with the GPT3 model, the GGZTD-P model showed greater accuracy improvement in the Arctic Ocean, Pacific Ocean, North America, South America, Africa, Europe, Oceania, and parts of Asia. Additionally, compared with the GPT3 model, the GGZTD-P model still shows some accuracy improvement in China. This improvement can be attributed to the fine detection of ZTD height intervals in the GGZTD-P model, allowing for a more accurate simulation of the vertical variations of ZTD across different height intervals. Therefore, the accuracy of the GGZTD-P model is improved to a certain extent in the China region with a large topographic relief.

**4.2 MERRA-2 data was used for verification**

The MERRA-2 atmospheric reanalysis data with 6-hour resolution in 2017 is also used as a reference value to validate the accuracy of the models. To assess the performance of the models, the global distribution of ZTD was divided into nine latitude regions, with each region covering a 20-degree interval. Then the bias and root-mean-square (RMS) error of the GGZTD-P and GPT3 models in different latitude intervals of the MERRA-2 profiled ZTD can be calculated. The results are presented in Fig. 9.

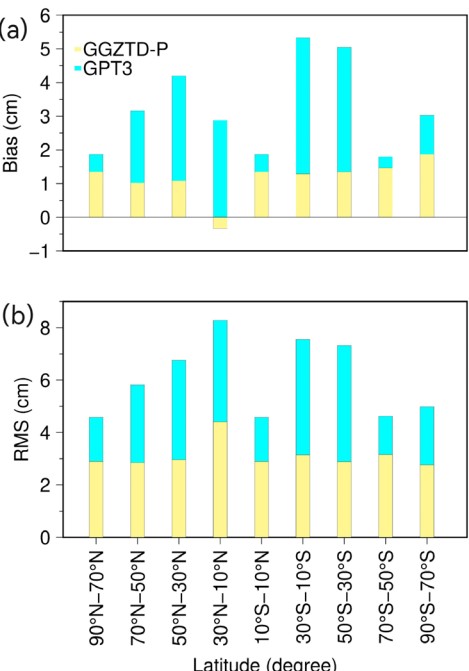

**Figure 9. Accuracy distribution of GGZTD-P model and GPT3 model in different latitude regions of MERRA-2 profile ZTD.**

Fig. 9 reveals that the GPT3 model has positive bias values in each latitude interval. Specifically, in the Northern hemisphere, the bias gradually increases with decreasing latitude, reaching a maximum of approximately -4.5 cm.

In the Southern hemisphere, the bias of the GPT3 model reaches the largest in 10°S to 30°S latitude interval. In contrast, the GGZTD-P model shows a small negative bias in the 10°S to 30°S latitude interval, but displays similar positive bias values to the GPT3 model in the other latitude ranges, indicating that the ZTD value calculated by the GGZTD-P model is larger than that of the radiosonde stations. The bias of the GGZTD-P model in the Northern Hemisphere does not show significant variations with decreasing latitude. In the Southern Hemisphere, however, the bias decreases with decreasing latitude. The GGZTD-P model exhibits smaller bias values in the low latitudes and higher bias values in the latitude range of 70°S to 90°S.

The RMS error of the GPT3 model increases with decreasing latitude in the Northern Hemisphere. In the Southern Hemisphere, however, it does not show a clear pattern with latitude, and its largest RMS error occurs in the latitude range of 10°N to 30°N. The GPT3 model exhibits the worst accuracy in the low-latitude region of the Northern hemisphere and better accuracy in the high-latitude region. In the Southern hemisphere, the accuracy is good at low and high latitudes but poor at mid-latitudes. In contrast, the RMS value of the GGZTD-P model shows a little variation with latitude in the Northern hemisphere, indicating that it is less affected by latitude factors, and its RMS error value is relatively stable. In the Southern hemisphere, the RMS value also shows no obvious variation with latitude.

Overall, the GGZTD-P model exhibits the largest errors in the range of 10°N to 30°N. Compared to the GPT3 model, the GGZTD-P model shows a greater improvement in accuracy in the low-latitude area, particularly in the latitude range of 10°S to 50°S, where it shows significant performance optimization. Although the GGZTD-P model also exhibits slight improvement in accuracy in the high-latitude area, it is not as pronounced as in the low-latitude area. Consequently, the GGZTD-P model demonstrates better ZTD correction performance globally than the GPT3 model.

**5 Global piecewise ZTD empirical grid model with different sliding window sizes**

In this study, we establish a combined empirical grid model through the integration of model coefficients obtained at various resolutions. The model parameters are optimized under the condition of low accuracy loss, which enhances the efficiency of the model. The surface parameters of GGZTD-P model and vertical profile parameters are combined to form three different models: GGZTD-P-1 with 1°×1° resolution for surface parameters and 2°×2° resolution for vertical profile parameters, GGZTD-P-2 with 1°×1° resolution for surface parameters and 5°×5° resolution for vertical profile parameters, and GGZTD-P-5 with 5°×5° resolution for both surface and vertical profile parameters. To evaluate the accuracy of the combined GGZTD-P model, the ZTD values of 545 radiosonde stations in 2017 and 2018 are served as reference values, and compared with that of 1°×1° grid resolutions of the GPT3 model. Statistical analysis of the accuracy of radiosonde stations profiled ZTD in each model is presented in Tab. 2, and Figs. 10 to 11 provide visual representation of the results.

Tab. 2 reveals that the accuracy of the combined GGZTD-P model decreases gradually as the resolution decreases. Nonetheless, the GGZTD-P-5 model still surpasses the GPT3 models in accuracy. In comparison to the GGZTD-P-1 and GGZTD-P-2 models, the RMS error of the GGZTD-P-5 model increased by 0.25 cm (8%) and 0.24 cm (7%), respectively. Conversely, when compared to the GPT3 models, the RMS errors of GGZTD-P-5 model decreased by

3.36 cm (49%). Additionally, the RMS error of the GGZTD-P-2 model increased by 0.44 cm (16%) relative to the
GGZTD-P-1 model.

335

**Table 2. Verify the combined GGZTD-P model and the GPT3 model in radiosonde profiled ZTD accuracy.**

| Model | Bias (cm) | | | RMS (cm) | | |
|---|---|---|---|---|---|---|
| | Max | Min | Mean | Max | Min | Mean |
| GGZTD-P-1 | 3.21 | -11.21 | 0.86 | 13.60 | 1.85 | 3.23 |
| GGZTD-P-2 | 3.21 | -11.23 | 0.87 | 13.61 | 1.86 | 3.24 |
| GGZTD-P-5 | 4.54 | -13.09 | 0.87 | 15.26 | 1.50 | 3.48 |
| GPT3 | 7.83 | -10.00 | 3.88 | 14.37 | 2.45 | 6.84 |

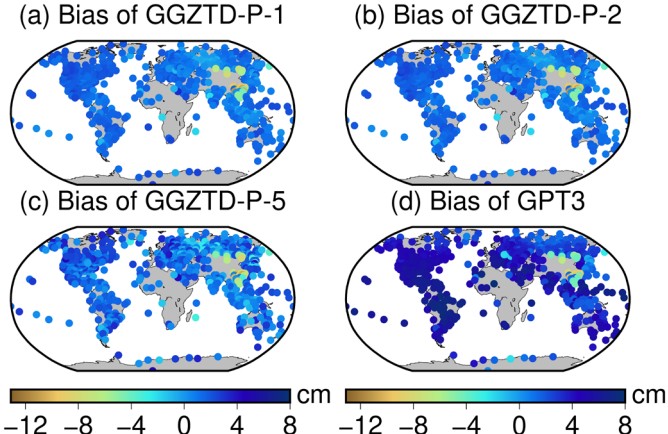

**Figure 10. Bias distribution of the GPT3 model and the combined GGZTD-P model in the global radiosonde profiled ZTD**
340    **accuracy.**

Figs. 10 and 11 reveal notable patterns in the ZTD values calculated by the combined GGZTD-P model. Overall,
the ZTD values obtained from the radiosonde stations display a positive bias on a global scale, indicating that the
ZTD values calculated by the combined GGZTD-P model tend to be higher than the ZTD values observed at the
345    radiosonde stations. However, in the Asian region, a significant negative bias is observed (It exhibits a higher bias
value when compared to other regions), suggesting that the ZTD values calculated by the combined GGZTD-P
model are consistently lower than the ZTD values from radiosonde stations. The combined GGZTD-P model and
the GPT3 model both show obvious negative bias values in the China region, indicating that the ZTD estimated by
the two models in China region is less than the ZTD value of the radiosonde station. It may be difficult to accurately
350    simulate the ZTD due to the complex and volatile climate and large terrain relief in the China region. Compared
with the GGZTD-P-1 model and the GGZTD-P-2 model, the GGZTD-P-5 model has a large bias in North America,
southern South America, Europe, Oceania and Antarctica. In terms of RMS error, the accuracy of GGZTD-P-5
model in parts of North America, parts of Europe and China is relatively poor compared with the GGZTD-P-1 model

and the GGZTD-P-2 model. The accuracy of GPT3 model in the global radiosonde station ZTD shows high accuracy in Antarctica and the Arctic Ocean. Compared with the GPT3 model, the combined GGZTD-P model shows a certain improvement in accuracy and has a better performance.

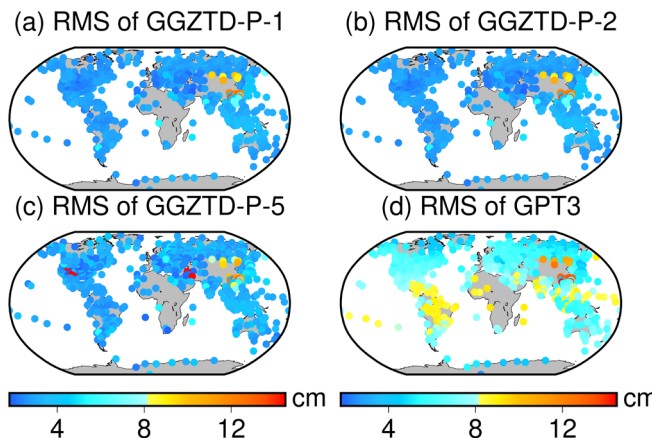

**Figure 11. RMS distribution of the GPT3 model and the combined GGZTD-P model in the global radiosonde profiled ZTD accuracy.**

## 6 Conclusion

ZTD is a critical parameter in GNSS positioning. Therefore, there is a pressing need to develop a new global ZTD model with high accuracy. In this study, we analyzed the temporal and spatial characteristics of global ERA5 reanalysis data and used FFT to analyze the periodicity of ZTD. The ZTD vertical profile data calculated from the ERA5 atmospheric reanalysis data are used to analyze the optimal elevation interval of ZTD. Then we propose GGZTD-P model based on piecewise exponential functions. The accuracy of the combined GGZTD-P model and GPT3 model is validated using data from radiosonde stations and MERRA-2. Using the profiled ZTD of radiosonde stations as the reference value, the GGZTD-P model had an RMS of 3.23 cm, which was 53% higher than that of the GPT3 model. When MERRA-2 reanalysis data was used as the reference value, GGZTD-P performed better than the GPT3 model in each latitude region. We also established empirical grid models with different window sizes using the sliding window algorithm. The models with different window sizes still showed relatively good performance, and users can choose an appropriate model based on their needs.

*Code and data availability.* Our work is available as a GitHub release at [pzaninelli/downloadERA5: Program to download ERA5 (github.com)](https://github.com). and on archive at CDO ([https://code.mpimet.mpg.de/projects/cdo/wiki](https://code.mpimet.mpg.de/projects/cdo/wiki)). All of the data generated during the current study and the code are available on ZENODO ([https://doi.org/10.5281/zenodo.8206173](https://doi.org/10.5281/zenodo.8206173)).

*Author contributions.* Liangke Huang: Conceptualization, Methodology, Formal analysis, Validation, Data curation, Writing – original draft, Writing – review & editing, Funding acquisition. Shengwei Lan: Conceptualization, Methodology, Formal analysis, Software, Validation, Data curation, Writing – original draft. Ge Zhu:

Conceptualization, Methodology, Formal analysis, Data curation, Writing – review & editing. Fade Chen: Validation, Investigation. Junyu Li: Investigation. Lilong Liu: Investigation, Funding acquisition.

*Competing interests.* The contact author has declared that none of the authors has any competing interests.

*Acknowledgments.* This work was funded by the National Natural Foundation of China (41704027), the Guangxi Natural Science Foundation of China (2023GXNSFAA026355), the Guangxi Key Laboratory of Spatial Information and Geomatics (19-050-11-24), the "Ba Gui Scholars" program of the provincial government of Guangxi, and Innovation Project of GuangXi Graduate Education (grant number: YCSW2023338). The authors would like to thank the University of Wyoming for providing the radiosonde profiles (http://www.weather.uwyo.edu/upperair/sounding.html). The reanalysis data, namely, the ERA5 and MERRA-2 products, are provided by the ECMWF (https://cds.climate.copernicus.eu/cdsapp#!/dataset/reanalysis-era5-single-levels?tab=form) and NASA (https://goldsmr4.gesdisc.eosdis.nasa.gov/data/MERRA2/), respectively.

*Financial support.* This research was funded by the National Natural Foundation of China (41704027), the Guangxi Natural Science Foundation of China (2023GXNSFAA026355), the Guangxi Key Laboratory of Spatial Information and Geomatics (19-050-11-24), the "Ba Gui Scholars" program of the provincial government of Guangxi, and Innovation Project of GuangXi Graduate Education (grant number: YCSW2023338), the State Key Laboratory of Geodesy and Earth's Dynamics, Innovation Academy for Precision Measurement Science and Technology, Chinese Academy of Sciences (SKLGED2023-3-1).

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
