# Peer review of "A global grid model for the estimation of zenith tropospheric delay considering the variations at different altitudes"

_Geoscientific Model Development, 2023_

## Author Comment (AC1)

This paper introduces an empirical model designed to estimate tropospheric delay at various altitudes, employing a set of complex modeling equations to express variations in ZTD. The model is developed based on ERA5 atmospheric reanalysis data, utilizing MERRA-2 and Radiosonde data as reference values. It demonstrates enhanced accuracy when compared to the GPT3 model across different spatiotemporal resolutions on a global scale. However, the manuscript still contains several issues that require attention and improvement. Here are my comments for enhancement:

Response: Thanks for your valuable comments and suggestions on our manuscript, which are very helpful for improving our manuscript. We have carefully revised our manuscript as suggested to meet the journal's requirements. The detailed revisions and responses are listed below:

L13, 'propose' should be corrected to 'proposed'.

Thank you for your suggestion. We have modified it **in L13** as follow:

"To address these limitations, we proposed a global piecewise ZTD empirical grid (GGZTD-P) model. This model considers the daily-cycle variation and latitude factor of ZTD, using the sliding window algorithm based on fifth-generation European Centre for Medium Range Weather Forecasts (ERA5) atmospheric reanalysis data."

In the introduction, it is advisable to cite recent articles that reflect the current state of the field. You may consider adding descriptions or references.
doi: 10.1007/s00190-022-01630-z
doi: 10.1007/s00190-021-01535-3

Based on your suggestion, we have added and substituted the following content in **L27 and L40**:

"Accurate Zenith Tropospheric Delay (ZTD) information can improve GNSS positioning precision (Nafisi et al., 2012; Zhang et al., 2021; Zhao et al., 2023a; Zhang et al., 2020; Zhou et al., 2021)."

"To overcome this limitation, researchers have developed several empirical models that do not rely on measured meteorological parameters (Li et al., 2020; Zhang et al., 2021)."

L45, the first occurrence of 'GPT' should explicitly mention its full name.

Thank you very much for your comments. We have modified it **in L46** as follow:

"The Global Pressure and Temperature (GPT) series models (Böhm et al., 2007; Lagler et al., 2013; Böhm et al., 2015; Landskron et al., 2018) are based on European medium-term prediction center (ECMWF) atmospheric reanalysis data and consider the temperature and pressure in cycles."

L80, what is the accuracy of radiosonde data, please also include some references.

Thank you for your suggestion. We cited some papers to show the accuracy of the data of the radiosonde station. We have modified it **in L88** as follows:

"Radiosonde data offers precise meteorological observations acquired through direct measurements. Zhao et al. (2019) found that ZTD derived from radiosonde is validated using GNSS data, with RMS errors of 19.1 mm. Shangguan et al. (2022) discovered that the bias and RMS of the ZTD data from 180 radiosonde stations compared with data from ERA5 worldwide were 8.5 mm and 13.2 mm, respectively."

Zhao, Q., Yao, Y., Yao, W., and Zhang, S.: GNSS-derived PWV and comparison with radiosonde and ECMWF ERA-Interim data over mainland China. Journal of Atmospheric and Solar-Terrestrial Physics, 182, 85-92. https://doi.org/10.1016/j.jastp.2018.11.004. 2019.

Shangguan, M., Cheng, X., Pan, X., Dang, M., Wu, L., and Xie, Z.: Assessments of global tropospheric delay retrieval from reanalysis based on GNSS data. Chinese Journal of Geophysics (in Chinese), 66(3), 939-950, https://doi.org/10.6038/cjg2022Q0023. 2023.

L98, clarify the unit of K3=375463 in line 98 of the manuscript; it appears to be a typographical error and should be K².

Thank you for your suggestion, this is our negligence. We have modified it **in L108** as follow:

"$k_1$ = 77.604K/Pa, $k_2$ = 64.79K/Pa, $k_2'$ = 22.97K/hPa and $k_3$ = 375463K²/hPa are all constant coefficients."

In section 3.3, the vertical correction grid model has a horizontal resolution of 2°×2°, but in section 3.4, the empirical grid model has a horizontal resolution of 1°×1°. Please clarify why empirical model and vertical profile model have different resolutions.

Thank you very much for your valuable comments. The rationale behind selecting varying resolutions is to finely optimize the model parameters, consequently enhancing the model's applicability with minimal loss of accuracy.

In Figure 3, it should be noted that the presence of a daily period variations cannot be conclusively demonstrated based on only the three grid points. At least select some grid points at the eastern hemisphere or at low latitude regions.

Thank you for your suggestion. We have modified it **in L135** as follows:

"To further confirm the daily period variations of ZTD, six ERA5 atmospheric reanalysis data grid points are selected randomly for on January 1, 2015. The results are presented in Fig. 3.

[Figure]

Figure 3. Time series of daily variations of ZTD.

Fig. 3 reveals that ZTD exhibits significant daily period variations in the six selected grid points, particularly at the grid points (0°, 90°E) and (1°S, 60°W) where significant daily period characteristics are observed. Thus, when constructing global ZTD models, it is important to consider daily period variations."

In Figure 7, the developed model is compared with the GPT3 model, while in Figure 9, the developed model is compared with the GPT3 model with different spatial resolution. I do not understand why you compare the results to the same reference model GPT3, but with different spatial resolution. As the results show, the GPT3-1 model has almost the same RMS with GPT3-5 model.

Your suggestion will help us a lot to improve our manuscript. We have deleted the GPT3-5 resolution results and only retain the GPT3-1 resolution results for comparison.

---

## Author Comment (AC3)

This paper establishes a global empirical ZTD model considering the variations at different altitudes. The quality of the presented materials is sufficient to be published in Geoscientific Model Development, although some changes are required as explained below.

Respone: I would like to thank you for taking the time and effort to go through us paper and providing constructive criticisms which are extremely valuable for us. I appreciate your thoughtful review and am grateful for your valuable insights.

Revise the sentence in line 54 of the manuscript regarding ZTD data from the radiosonde station to specify the correct data source and avoid confusion, as radiosonde stations do not provide ZTD data directly.

We are grateful for the suggestion, and realize that the description here is not accurate enough. We have corrected it **in L66** as follow:

"The accuracy of the GGZTD-P model was evaluated by comparing it with profiled ZTD data from 545 radiosonde stations in 2017 and 2018, as well as the Modern-Era Retrospective analysis for Research and Applications, Version 2 (MERRA-2) atmospheric reanalysis data from 2017. It should be explained that the ZTD data of radiosonde and MERRA-2 is calculated by integration."

In section 3.4, please provide a more detailed introduction of the GGZTD-P model, such as a description of the data used for its establishment. This would enhance the reader's understanding, as data specifics are crucial.

We agree with the comment and re-wrote the sentence in the revised manuscript in **L202** as follow:

"ERA5 atmospheric reanalysis data ZTD on the surface will be uniformly converted to the position of the sliding window's average elevation. This conversion is based on the piecewise global ZTD vertical profile model, GZTD-P model, taking into account the elevation position of each window. The model is based on the ZTD values at the sliding window's average elevation. Utilizing the GZTD-P model, ZTD data for all window from 2012 to 2016 were vertically interpolated to calculate the ZTD value at the average elevation of each window after correction. The detailed process is shown in Fig. 7. To estimate the coefficients in each window, the least-squares adjustment is utilized, considering the annual, semi-annual, daily, and semi-daily variations, as well as the latitude factor. Finally, the global ZTD empirical grid model (GGZTD-P) is developed based on a piecewise expression, with a resolution of 1°×1°. The model can be expressed as follows:"

The full names of MERRA-2 and ZTD, mentioned in lines 69 and 86, were previously indicated when they first appeared in the preceding paragraph and need not be reemphasized.

Thank you very much for your comments. We'd like to apologize again for our carelessness, and we have changed it **in L77 and L95** as follow:

"MERRA-2 is a state-of-the-art atmospheric reanalysis dataset developed by NASA (Chen et al., 2019; Huang et al., 2022; Randles et al.,2017)."

"Atmospheric reanalysis data can provide meteorological parameters according to standard atmospheric pressure profiles. Integration method is used to calculate the ZTD."

P9, in section 4, Accuracy verification, the authors at least need to provide an explanation of how the GPT3 model was developed and disclose the data utilized in its formation. This information is crucial, especially considering the extensive use of the GPT3 model as a reference throughout the manuscript to assess the performance of their novel model estimates.

We appreciate for your effort to review our manuscript. In the introduction, we have described the development process of GPT3 model. In order to avoid repetition, we have added what data the GPT3 model is based on.We have changed it **in L225** as follow:

"In order to verify the stability of the established model in the global region, two sets of data are used as reference values and compared with the GPT3 model. The GPT3 model was developed utilizing a 15-year dataset of monthly average ERA-Interim profiles. Currently, it functions as a highly accurate tropospheric model.

$$M = S_0 + S_1 \cdot \cos\left(2\pi\frac{DOY}{365.25}\right) + S_2 \cdot \sin\left(2\pi\frac{DOY}{365.25}\right)$$
$$+ S_3 \cdot \cos\left(4\pi\frac{DOY}{365.25}\right) + S_4 \cdot \sin\left(4\pi\frac{DOY}{365.25}\right)$$
$$\tag{9}$$

In Eq. (9), $M$ represents the tropospheric meteorological parameters (temperature, water vapor pressure, specific humidity, etc), and $S_i$ represents the annual mean value, annual and, semi-annual period coefficients. The Saastamoinen model and the Askne model were adopted to compute zenith hydrostatic delay (ZHD) and zenith wet delay (ZWD) with the obtained meteorological parameters.

$$ZHD = \frac{0.0022768P}{1 - 0.00266\cos(2\theta) - 0.00000028h} \tag{10}$$

$$ZWD = 10^{-6}(k_2' + \frac{k_3}{T_m}) \cdot \frac{R_d}{(\lambda + 1) \cdot g_m} \cdot e \tag{11}$$

In Eqs. (10) and (11), $P$ stands for pressure, $\theta$ stands for latitude, $h$ stands for elevation, $g_m$ is the average acceleration of gravity, $\lambda$ stands for the drop factor of water vapor pressure, $T_m$ stands for the atmospheric weighted mean temperature, and $k_2'$ = 22.97K/hPa, $k_3$ = 375463K$^2$/hPa, $R_d$ = 287.054J/kg$\cdot$K are all constant coefficients."

Please ensure consistency in the naming format for figures and tables throughout the document, such as "Figure.1" and "Figure 1. "

We are grateful for the suggestion. After consulting multiple published papers in this journal, we unified the format used, as shown in Fig. 1. It has been amended in the whole manuscript.

In line 311, how do you define the term "significant bias"? Is a significant bias, in your view, characterized by a statistically significant difference from an expected value, as determined through statistical testing?

Thank you very much for your comments. We have neglected the problem. In fact, what we aim to illustrate is that this region exhibits a higher bias value compared to others. We have changed it **in L345** as follow:

"However, in the Asian region, a significant negative bias is observed (It exhibits a higher bias value when compared to other regions), suggesting that the ZTD values calculated by the combined GGZTD-P model are consistently lower than the ZTD values from radiosonde stations."

---

## Author Comment (AC4)

This manuscript introduces a globally empirical ZTD model using ERA5 atmospheric reanalysis data. It offers a well-structured analysis of temporal and spatial characteristics, presenting intriguing research within the domain of high-precision tropospheric modeling. However, I have identified several minor issues that warrant attention and correction. Therefore, in preparation for potential publication, I recommend implementing the following modifications:

Respone: According to the your comments, we have revised the manuscript. If there are any other modifications we could make, we would like very much to modify them and we really appreciate your help. The detailed revisions and responses are listed below:

1. In the introduction, the authors introduce only classic models. It is suggested to supplement the literature with recent global ZTD empirical models.

Thank you very much for your comments. we have substituted and added the following content **to L27 and L55**:

"Accurate Zenith Tropospheric Delay (ZTD) information can improve GNSS positioning precision (Nafisi et al., 2012; Zhang et al., 2022; Zhao et al., 2023a; Zhang et al., 2020; Zhou et al., 2021)."

"Furthermore, Yang et al. (2021) employed an Artificial Neural Network (ANN) to effectively mitigate the systematic deviation within the GPT3 model, leading to improved ZTD accuracy in Hong Kong, China. Zhao et al. (2023) took into account the residual term between the GPT3 model and GNSS observations ZTD to develop a novel model specific to China (CHZ). Additionally, Li et al. (2023) discover the disparities between ERA5 and GNSS-based ZTD, prompting the creation of a new global model (IGGZTD-S). This new model demonstrated exceptional performance in Precise Point Positioning (PPP), particularly in the vertical direction."

Zhao, Q., Liu, K., Sun, T., Yao, Y., and Li, Z.: A novel regional drought monitoring method using GNSS-derived ZTD and precipitation. Remote Sensing of Environment, 297, 113778. https://doi.org/10.1016/j.rse.2023.113778. 2023a.

Yang, F., Guo, J., Zhang, C., Li, Y., and Li, J.: A Regional Zenith Tropospheric Delay (ZTD) Model Based on GPT3 and ANN. Remote Sensing, 13, 838. https://doi.org/10.3390/rs13050838. 2021.

Zhao, Q., Su, J., Xu, C., Yao, Y., Zhang, J., and Wu, J.: High-precision ZTD model of altitude-related correction. IEEE Journal of Selected Topics in Applied Earth Observations and Remote Sensing, 16, 609-621. https://doi: 10.1109/JSTARS.2022.3228917. 2023b.

Li, H., Zhu, G., Kang, Q. and Wang, H.: A global zenith tropospheric delay model with ERA5 and GNSS-based ZTD difference correction. GPS Solutions, 27, 154. https://doi.org/10.1007/s10291-023-01503-8. 2023.

2. In 158, "This may be due to the complex climate variations in these areas causing more dramatic ZTD variations ", are the authors sure that semi-daily period amplitude is mainly due to complex climate variations? Is there any other explanation for this phenomenon?

Thank you for your suggestion. We realize that this statement may be too simplistic, We have modified it **in L169**:

"This may be due to the fact that these regions are located at the junction of the ocean and land and are in the same direction as the northeast (Northern Hemisphere) and southeast (Southern hemisphere) equatorial trade winds (Yao et al., 2013), indicating that the distribution of ZTD is not only related to meteorological variables and topography, but also influenced by thermodynamic circulation (Yao et al., 2013)."

Yao, Y., Zhu, S. and Yue, S.: A globally applicable, season-specific model for estimating the weighted mean temperature of the atmosphere. Journal of Geodesy, (86), 1125–1135. https://doi.org/10.1007/s00190-012-0568-1. 2012.

Yao, Y., He, C., Zhang, B., and Xv, C.: A new global zenith tropospheric delay model GZTD. Chinese Journal of Geophysics, 56(7), 2218-2227. https://doi:10.6038/cjg20130709. 2013.

3. In 162, please include specific references to substantiate the description and enhance its credibility.

Thank you for your suggestion. We have revised it in **L173**:

"According to relevant studies, ZTD values are primarily associated with latitude factors on a global scale, while showing a smaller correlation with longitude factors (Chen et al., 2020; Huang et al., 2022)."

Chen, P., Ma, Y., Liu, H., and Zheng, N.: A new global tropospheric delay model considering the spatiotemporal variation characteristics of ZTD with altitude coefficient. Earth and Space Science, 2020, 7(4), e2019EA000888. https://doi.org/10.1029/2019EA000888. 2020.

Huang, L., Zhu, G., Peng, H., Liu, L., Ren, C., and Jiang, W.: An improved global grid model for calibrating zenith tropospheric delay for GNSS applications. GPS Solutions, 27(1), 17. https://doi.org/10.1007/s10291-022-01354-9. 2023.

4. In 183, consider providing a more comprehensive explanation of "Hs" to ensure clarity and understanding.

Thank you very much for your comments. We have modified it in **L197** as follow:

"In Eqs. (4) and (5), $H_s$ stands for ZTD value at the average elevation, $H_t$ stands for target elevation, $H_r$ stands for reference elevation, and $ZTD_t$ stands for ZTD value at target elevation. $a_i$ stands for the constant, annual and semi-annual period correction factor. $ZTD_{r1}$, $ZTD_{r2}$, $ZTD_{r3}$, $ZTD_{r4}$ stands for ZTD values at the reference elevations of different piecewise, respectively."

5. In 197, please correct the error in the expression of "ai".

We appreciate for your effort to review our manuscript. As you said, it was a mistake, We have revised it **in L215** as follow:

"$a_i$ represents the constant, latitude, annual and semi-annual period correction factor, $H_t$ stands for target elevation, $\varphi$ represents latitude, $DOY$ represents year day, $HOD$ represents time."

6. The authors need to pay attention to the format of all the images in the full manuscript, some partitions have subtitles, some do not, need to be uniform.

Thank you very much for your comments. It was an oversight on our manuscript, and we modified and unified the images:

[Figure]

**Figure 8. ZTD profiled accuracy distribution of GGZTD-P model and GPT3 model for global radiosonde stations in 2017 and 2018.**

[Figure]

(a) Bias of GGZTD-P-1    (b) Bias of GGZTD-P-2

(c) Bias of GGZTD-P-5    (d) Bias of GPT3-1

**Figure 10. Bias distribution of the GPT3 model and the combined GGZTD-P model in the global radiosonde profiled ZTD accuracy.**

[Figure]

(a) RMS of GGZTD-P-1    (b) RMS of GGZTD-P-2

(c) RMS of GGZTD-P-5    (d) RMS of GPT3-1

**Figure 11. RMS distribution of the GPT3 model and the combined GGZTD-P model in the global radiosonde profiled ZTD accuracy.**

7. In Figure 6, can the authors explain why they chose a height of 6km and not some other height?

Thank you very much for your comments. We chose 6km because the highest global surface height provided by ERA5 is around 6km. Therefore, 6 km was chosen to analyze the variation of ZTD. In addition, Figure 6 proves that the global distribution of ZTD of ERA5 atmospheric reanalysis data is closely related to latitude factors.

8. In Figure 8, the author needs to pay attention to the border of each small bar chart.

We are grateful for the suggestion. We have corrected it as follow:

[Figure]

**Figure 9. Accuracy distribution of GGZTD-P model and GPT3 model in different latitude regions of MERRA-2 profile ZTD.**

---

## Author Comment (AC5)

A global empirical model was designed in this paper to estimate the zenith troposphere delay in different altitudes. The ERA5 reanalysis data, MERRA-2 and other data were applied for validation. However, some more details are need improving the manuscript before the possible publication:

Response: Thank you for your letter and for the reviewers comments 'concerning our manuscript. Those comments are all valuable and very helpful for revising and improving our paper, as well as the important guiding significance to our researches. We have studied comments carefully and have made correction which we hope meet with approval. The main corrections in the paper and the responds to the reviewer's comments are as flowing:

1. in Line 158: "This may be due to the complex climate variations in these areas causing more dramatic ZTD variations." authors please provide more supplement to prove that the "complex climate variations" are the caution of the ZTD. Besides, when we talk about the climate, we usually focus on the statistical state of the regional or global weather characteristics, such as the monthly/annual mean the air temperature, wind speed/direction, humidity, ...... and also the extreme weather conditions in a certain region over a period. However, in my understanding, the weather variables that affect ZTD may be more in their immediate state than in their climatic state, so using "meteorological variables" seems more reasonable in this part of discussion.

Thank you very much for your comments. It turns out that our description of the complex changes in ZTD is too simplistic, and after a careful review of the literature, we find that this is due to a number of factors, For example, "meteorological variables", " topography", "thermodynamic circulation"...

We have revised the description and added references **in L169**:

"This may be due to the fact that these regions are located at the junction of the ocean and land and are in the same direction as the northeast (Northern Hemisphere) and southeast (Southern hemisphere) equatorial trade winds (Yao et al., 2012), indicating that the distribution of ZTD is not only related to meteorological variables and topography, but also influenced by thermodynamic circulation (Yao et al., 2013)."

Yao, Y., Zhu, S. and Yue, S.: A globally applicable, season-specific model for estimating the weighted mean temperature of the atmosphere. Journal of Geodesy, (86), 1125–1135. https://doi.org/10.1007/s00190-012-0568-1. 2012.

Yao, Y., He, C., Zhang, B., and Xv, C.: A new global zenith tropospheric delay model GZTD. Chinese Journal of Geophysics, 56(7), 2218-2227. https://doi:10.6038/cjg20130709. 2013.

2. the equations in the GZTD-P and GGZTD-P are provided in the section 3.3 and 3.4. Although both of them are empirical model, more introduction of the physical images and model explanations still need to be provided to us, either in the main draft or in the supplement. Otherwise, as an article on model development, it may be difficult for us to better understand the basis of those eqs 1-8.

Your suggestion will help us a lot to improve our manuscript. In order to make it easier for all to understand, we have added a flow chart of the model, and give a more detailed description. We have modified it **in L202**:

"ERA5 atmospheric reanalysis data ZTD on the surface will be uniformly converted to the position of the sliding window's average elevation. This conversion is based on the piecewise global ZTD vertical profile model, GZTD-P model, taking into account the elevation position of each window. The model is based on the ZTD values at the sliding window's average elevation. Utilizing the GZTD-P model, ZTD data for all window from 2012 to 2016 were vertically interpolated to calculate the ZTD value at the average elevation of each window after correction. The detailed process is shown in Fig. 7. To estimate the coefficients in each window, the least-squares adjustment is utilized, considering the annual, semi-annual, daily, and semi-daily variations, as well as the latitude factor. Finally, the global ZTD empirical grid model (GGZTD-P) is developed based on a piecewise expression, with a resolution of 1°×1°. The model can be expressed as follows:

$$ZTD_t = \begin{cases} ZTD_r * \exp\left(H_{s1} * \left(H_t - \mathrm{H}_r\right)\right)\left(H_t < 3\mathrm{km}\right) \\ ZTD_3 * \exp\left(H_{s2} * \left(H_t - 3\right)\right)\left(3\mathrm{km} \le H_t < 8\mathrm{km}\right) \\ ZTD_8 * \exp\left(H_{s3} * \left(H_t - 8\right)\right)\left(8\mathrm{km} \le H_t < 16\mathrm{km}\right) \\ ZTD_{16} * \exp\left(H_{s4} * \left(H_t - 16\right)\right)\left(H_t \ge 16\mathrm{km}\right) \end{cases}$$

(6)

$$MP = A_0 + A_1 \cdot \cos\left(2\pi \frac{HOD}{24}\right) + A_2 \cdot \sin\left(2\pi \frac{HOD}{24}\right)$$
$$+ A_3 \cdot \cos\left(4\pi \frac{HOD}{24}\right) + A_4 \cdot \sin\left(4\pi \frac{HOD}{24}\right)$$

(7)

$$A_i = \alpha_1 + \alpha_2 \cdot \varphi + \alpha_3 \cdot \cos\left(2\pi \frac{DOY}{365.25}\right) + \alpha_4 \cdot \sin\left(2\pi \frac{DOY}{365.25}\right)$$
$$+ \alpha_5 \cdot \cos\left(4\pi \frac{DOY}{365.25}\right) + \alpha_6 \cdot \sin\left(4\pi \frac{DOY}{365.25}\right)$$

(8)

In Eqs. (6) (7) and (8), $MP$ stands for the ZTD value at the average elevation, 3 km elevation, 8 km elevation and 16 km elevation, and $A_i$ stands for the daily period coefficient. $a_i$ stands for the constant, latitude, annual and semi-annual period correction factor, $\varphi$ stands for latitude, $DOY$ stands for year day, $HOD$ stands for time."

[Figure]

Figure 7. Flowchart depicting the development and use of the model.